

# Advances in understanding the role of lncRNA in ferroptosis

Yating Wen, Wenbo Lei, Jie Zhang, Qiong Liu and Zhongyu Li

Pathogenic Biology Institute, Hengyang Medical College, University of South China, Hengyang, Hunan, China

## ABSTRACT

LncRNA is a type of transcript with a length exceeding 200 nucleotides, which was once considered junk transcript with no biological function during the transcription process. In recent years, lncRNA has been shown to act as an important regulatory factor at multiple levels of gene expression, affecting various programmed cell death modes including ferroptosis. Ferroptosis, as a new form of programmed cell death, is characterized by a deficiency of cysteine or inactivation of glutathione peroxidase, leading to depletion of glutathione, aggregation of iron ions, and lipid peroxidation. These processes are influenced by many physiological processes, such as the Nrf2 pathway, autophagy, p53 pathway and so on. An increasing number of studies have shown that lncRNA can block the expression of specific molecules through decoy effect, guide specific proteins to function, or promote interactions between molecules as scaffolds. These modes of action regulate the expression of key factors in iron metabolism, lipid metabolism, and antioxidant metabolism through epigenetic or genetic regulation, thereby regulating the process of ferroptosis. In this review, we snapshotted the regulatory mechanism of ferroptosis as an example, emphasizing the regulation of lncRNA on these pathways, thereby helping to fully understand the evolution of ferroptosis in cell fate.

## WHY THIS REVIEW IS NEEDED AND WHO IT IS INTENDED FOR

LncRNA used to be considered as transcriptional noise. With the development of molecular biology, the regulatory role of lncRNA was identified. LncRNA affects various physiological processes such as cell differentiation and cell death at multiple gene expression levels. Ferroptosis, as a rapidly advancing new form of death in research over the past decade, has continuously improved its specific molecular mechanisms. Ferroptosis has been proven to be caused by different pathological conditions and is closely related to the occurrence and development of many diseases. Growing studies have shown that lncRNA seems to be able to prevent tumor occurrence by regulating ferroptosis. Although the roles of lncRNA and ferroptosis in different diseases seem complex, the fundamental mechanism is to affect the expression of key factors in ferroptosis, which can be beneficial or harmful to the body. Therefore, this review summarizes and discusses the regulation of lncRNA in the process of ferroptosis, clarifying the essential lncRNA in ferroptosis and its

Corresponding author
Zhongyu Li, lzhy1023@hotmail.com

related pathways. Our review will attract pharmacists, biological scientists, and clinical workers engaged in ncRNA and ferroptosis research to bridge the perspectives of the two fields, and also enable researchers interested in life sciences to fully understand them.

## SURVEY METHODOLOGY

Primary and secondary literature related to the topics of this review were evaluated by searching on PubMed and Web of Science. The following terms were used individually or in combination for search: lncRNA, ferroptosis, iron metabolism, lipid metabolism, glutathione metabolism, Nrf2, autophagy, ferritinophagy, mitophagy, lipophagy, p53. A preliminary screening of the identified articles was conducted to ensure they were relevant to the topic. Bioinformatics analysis articles unverified by experiment, and studies that do not involve molecular mechanisms are excluded from the search results.

## INTRODUCTION

LncRNA is a type of transcript with a length of over 200 nucleotides, and its variety and number are abundant (*Mattick et al., 2023*). Initially, it was considered to be a junk transcript without biological function (*Frankish et al., 2023*). However, within the past decade, the role of lncRNAs in the regulation of biological function was discovered (*Kopp & Mendell, 2018*). In fact, many studies have shown that lncRNA acts as a crucial regulatory factor at multiple levels of gene expression, including transcription, post-transcription, and translation. As a result, it impacts physiological processes such as cell differentiation, cell cycle, and cell death, *etc.* (*Wen et al., 2020*). LncRNAs are involved in the occurrence and development of many diseases by these regulatory mechanisms, especially in the monitoring of cell death (*Alammari et al., 2024*; *Huang et al., 2020*).

Cell death encompasses various forms, which can be divided into accidental and regulatory cell death based on functional differences (*Galluzzi et al., 2018*). Accidental cell death is triggered by accidental injury stimuli that exceed the adjustable capacity of cells. Regulatory cell death is associated with signaling cascades involving effector molecules, with unique biochemical, morphological, and immunological consequences, such as necrosis, pyroptosis, lysosomal-dependent cell death, and ferroptosis, *etc.*, (*Galluzzi et al., 2018*). Increasing evidence implies that lncRNA is closely related to ferroptosis (*Xie & Guo, 2021*).

Ferroptosis was first proposed by *Dixon et al. (2012)* in 2012, characterized by a large iron-dependent accumulation of lipid ROS. The initial characteristics of ferroptosis are the loss of cysteine or the inactivation of GPX4, leading to the depletion of glutathione (GSH), iron aggregation, and lipid peroxidation (*Dixon, 2017*). The accumulation of iron ions causes specific morphological changes. Ferroptotic cells lack apoptotic morphological features (such as cell shrinkage and membrane foaming; *Conrad et al., 2016*), and often exhibit significant changes in mitochondrial ultrastructure under electron microscopy, such as changes in mitochondrial morphology and ridge structure (*Conrad et al., 2016*). After ferroptosis induction, the volume of mitochondria decreases, the number of mitochondrial cristae decreases or is missing (*Dixon et al., 2012*), the density of mitochondrial membranes increases (*Yagoda et al., 2007*), mitochondrial membrane

potential increases, and mitochondrial outer membrane damage (*Friedmann Angeli et al., 2014*). Athough ferroptosis inducers cause oxidative damage to cell DNA, the morphology of the nucleus is normal, and chromatin does not condense, indicating a significant difference between ferroptosis and apoptosis (*Song et al., 2016*). At the genetic level, it manifests as an imbalance of molecules in iron, lipid, and antioxidant metabolism processes, such as SLC7A11, GPX4, TfR1, ACSL4, *etc.*, (*Desideri, Ciccarone & Ciriolo, 2019*; *Manz et al., 2016*) (see below).

Numerous studies have demonstrated the significant role of ferroptosis in killing tumor cells and inhibiting tumor growth (*Hassannia, Vandenabeele & Vanden Berghe, 2019*). Several tumor suppressors have been proven to make cells sensitive to ferroptosis, such as p53, BRCA1-associated protein 1 (BAP1), Kelch like ECH-associated protein 1 (KEAP1) and so on (*Fan et al., 2017*; *Jiang et al., 2015*; *Zhang et al., 2018b*). Ferroptosis has been identified as one of the important causes of tumor cell death, particularly in cases of pancreatic cancer, breast cancer, and non-small cell lung cancer (*Ma et al., 2016*; *Wang et al., 2024b*; *Yamaguchi, Kasukabe & Kumakura, 2018*). However, ferroptosis may also play a role in development of cancer in the context of tumor immunity (*Matsushita et al., 2015*; *Yang et al., 2014*). Ferroptosis has been the major focus of cancer treatment research to date. This article provides an overview of the function of lncRNA in ferroptosis.

## THE FUNCTIONS AND CLASSIFICATIONS OF LNCRNA

Similar to mRNA, lncRNA is transcribed by RNA polymerase II (RNAP-II) and undergoes termination, polyadenylation, and splicing (*Mattick et al., 2023*; *Rinn & Chang, 2020*). According to their genomic location, lncRNAs can be categorized into five types: intergenic, intronic, sense, antisense and bidirectional lncRNA. Their functions are typically classified into three categories based on their mechanisms at target genes: decoy, guide, and scaffold. In recent years, some lncRNAs have also been found to possess coding functions (*Mattick et al., 2023*). Here, we briefly introduced several functions of lncRNAs as follows.

### Decoy

LncRNAs plays a significant role by acting as baits to block the expression of specific effectors. These lncRNAs can directly bind to certain molecules such as transcriptional regulatory factors or chromatin folding proteins, thereby preventing the protein from functioning (*McFadden & Hargrove, 2016*; *Zhang et al., 2018a*). LncRNA mainly serve as baits in two ways. Firstly, lncRNA can block transcription factor function by directly binding to transcription factors, thereby inhibiting downstream gene expression (Fig. 1A). For instance, in low levels of DNA damage, lncRNA PAND directly binds to NF-YA, a nuclear transcription factor, and prevents the expression of apoptotic genes during DNA damage (*Hung et al., 2011*). Secondly, lncRNAs can also act as sponges to absorb microRNAs (miRNAs), and prevent miRNAs from targeting and binding to mRNA, thereby upregulating the expression of target genes and playing a competitive endogenous RNA (ceRNA) role (Fig. 1B) (*Majumdar et al., 2024*; *Sengupta et al., 2024*). For example, lncRNA MT1JP is associated with gastric cancer (GC) survival. MT1JP adsorbs

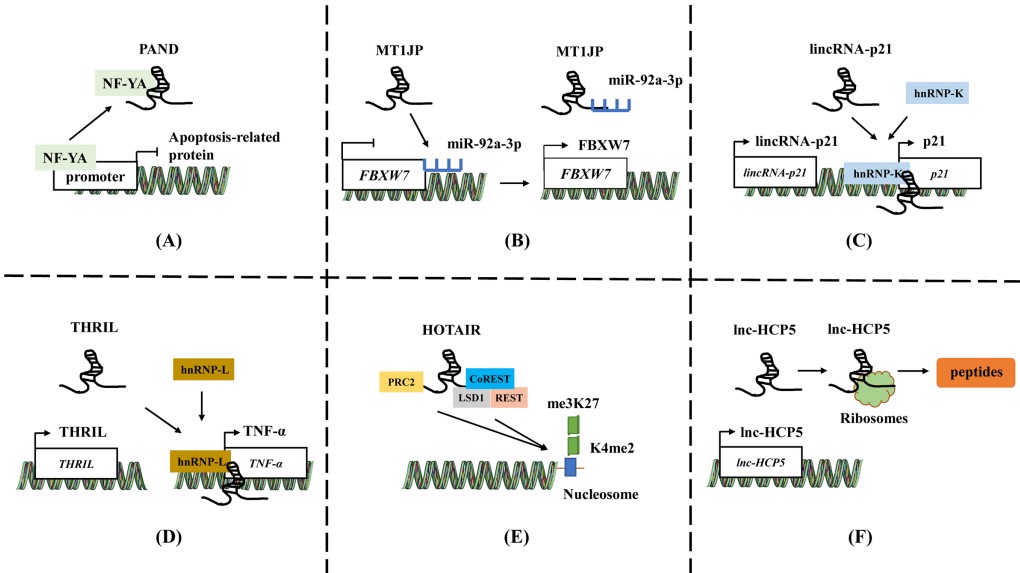

**Figure 1 The general mechanisms of lncRNAs.** (A) LncRNA PAND directly binds to NF-YA and prevents the expression of apoptotic genes. (B) LncRNA MT1JP adsorb miR-92a-3p to target and regulate the expression of FBXW7. (C) LincRNA-p21 activates the expression of neighboring gene *p21* by recruiting and binding to hnRNP-K. (D) LncRNA THRIL forms a complex with hnRNA-L to bind to the promoter of TNF-α, increasing the TNF-α expression. (E) HOTAIR binds to polycomb repressive complex 2 (PRC2) through 5′-UTR, while 3′-UTR binds to the LSD1/CoREST/REST complex. (F) The ORF in lncRNA HCP5 encodes a protein with 132 amino acids. Template of nucleic acid element come from SMART (https://smart.servier.com/smart_image/dna/).

miR-92a-3p to target and regulate the expression of FBXW7, which is a tumor suppressor molecule that plays an important role in tumor apoptosis, epithelial cell differentiation, and drug resistance in GC (*Li et al., 2016*).

## Guide

LncRNAs act as guiding molecules that direct specific proteins to their target locations and carry out biological functions. They often interact with transcription factors or RNA-binding proteins (RBPs), which recognize specific gene sequences and regulate gene transcription (*Chen et al., 2016*; *Zhang et al., 2018a*). The impact of lncRNA is often achieved through either *cis* or *trans* interactions. *Cis* action means that lncRNA affects the expression of genes located on the same chromosome. LncRNAs located upstream and downstream of coding proteins may intersect with other *cis*-acting elements of promoters or co-expressed genes, thereby regulating gene expression at the transcriptional or post-transcriptional level (Fig. 1C). For instance, lincRNA-p21 activates the expression of its neighboring gene *p21* and collaborates with hnRNP-K to co-activate p21 transcription. (*Ferrer & Dimitrova, 2024*; *Winkler et al., 2022*). Conversely, *trans* action involves lncRNAs leaving the transcription site after being transcribed and targeting distant transcription activating factors or repressors to regulate genome expression or affect gene localization in cells (Fig. 1D) (*Gil & Ulitsky, 2020*; *Kopp & Mendell, 2018*). LncRNA THRIL forms a complex with hnRNA-L to bind to the promoter of TNF-α, thereby upregulating

the TNF-α expression (*Li et al., 2014*). This mechanism allows lncRNAs to play a role in the precise regulation of intracellular signaling pathway networks and participate widely in various cellular physiological processes.

## Scaffold

In addition to the two functions mentioned above, lncRNA can also act as a central platform, promoting interactions between various molecules and proteins. The scaffold properties allow for the assembly of different types of macromolecular complexes, and it facilitates the aggregation and integration of information from different signaling pathways (*Jiang et al., 2018*; *Schmidt et al., 2019*). HOTAIR binds to polycomb repressive complex 2 (PRC2) through 5′-UTR, while 3′-UTR binds to the LSD1/CoREST/REST complex, providing a scaffold for the binding of two different complexes (Fig. 1E). The role enables the successful assembly of RNA-mediated PRC2 and LSD1, connecting histone methylase and demethylase, and coordinating their targeted action on chromatin (*Tsai et al., 2010*).

## Translation

Previous studies have shown that a minority of lncRNAs have an open reading frame (ORF) and may possess coding ability (*Huang et al., 2017*; *Matsumoto et al., 2017*). The micropeptides translated from lncRNA can play a regulatory role in mitochondrial metabolism, tumor development, and other processes by regulating related signaling pathways, inhibiting metabolic reprogramming, *etc.*, (Fig. 1F) (*Huang et al., 2017*; *Zheng et al., 2023b*; *Zhou et al., 2023*). The ORF in lncRNA HCP5 encodes a protein with 132 amino acids, which can regulate ferroptosis and promote the progression of TNBC by regulating glutathione peroxidase 4 (GPX4) expression and lipid ROS levels (*Tong et al., 2023a*). Tumor-related proteins or peptides encoded by lncRNA may be used in combination with conventional anticancer drugs or chemotherapy to enhance treatment efficacy and reduce mortality. For example, the ASRPS peptide encoded by lncRNA is essential in the malignant progression of TNBC (*Wu et al., 2020*), while the HOXB-AS3 peptide inhibits the of colorectal cancer (*Wang et al., 2020*). *Wu et al. (2020)* summarized and reviewed the lncRNA and circRNA encoding peptides in tumors.

# LNCRNA REGULATES THE PROCESS OF FERROPTOSIS

## LncRNA and iron metabolism

Iron is an essential trace element that plays a role in systemic oxygen transport and serves as an electron donor or acceptor in many biological functions. It is essential for maintaining cellular metabolism and important physiological activities (*Muchowska, Varma & Moran, 2019*). If cells are overloaded with iron, the free ferrous ions have strong oxidizing properties and are prone to Fenton reaction with $H_2O_2$, producing hydroxyl radicals that can cause oxidative damage to DNA, proteins, and membrane lipids. This promotes lipid peroxidation reactions and damages cell membranes, ultimately leading to cell death (Fig. 2A) (*Wang et al., 2024a*).

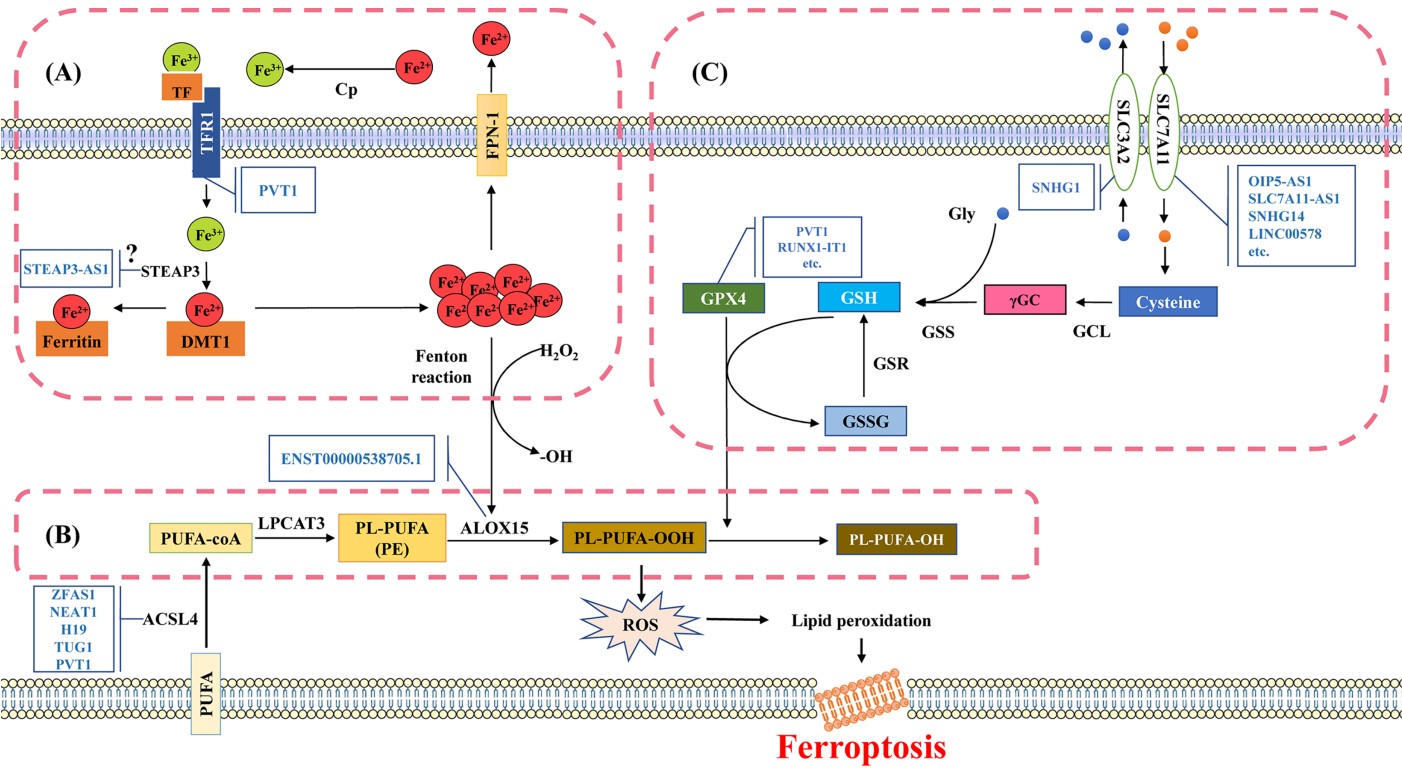

**Figure 2 Molecular mechanisms and signaling pathways of ferroptosis.** (A) Mechanisms of iron metabolism in ferroptosis. Iron in the circulatory system binds to TF. During the uptake, TF binds to TfR1 on the cell membrane and then internalized into the cell. $Fe^{3+}$ is released from TF and reduced to divalent iron ions through iron reductase, which binds to DMT1 and is transported to the cytoplasm. The divalent iron in the cytoplasm enters the transient cytosolic LIP. LIP is a chelating and redox-active iron pool that cells use for various metabolic processes or store in ferritin, the main iron storage protein in cells. Most of the iron in ferritin is in the ferrite state ($Fe^{3+}$). Excess divalent iron is transported from cells to the circulation through FPN-1, and then oxidized by CP to trivalent iron, which binds to TF in the serum. (B) Mechanisms of lipid metabolism in ferroptosis. The PUFA substrates AA and AdA are activated by ACSL4 to produce AA-CoA and AdA-CoA. Subsequently, AA/AdA-CoA is esterified by LPCAT3 to form PE-AA/AdA. PE-AA/AdA is oxidized by ALOX15 to a cytotoxic lipid hydroperoxide PE-AA/AdA-OOH. (C) Mechanisms of glutathione metabolism in ferroptosis. The synthesis process of glutathione is mainly divided into two steps. Firstly, the cells absorb cystine from the extracellular space and output glutamate through the System Xc- in a 1:1 ratio. System Xc- is an amino acid transport system widely distributed in the phospholipid bilayer and is a component of the antioxidant system in cells, consisting of two subunits (SLC7A11 and SLC3A2). Cystine is converted into cysteine in the cytoplasm through a reduction reaction that consumes NADPH, and forms γGC under the action of GCL; Subsequently, GSS catalyzes the production of glutathione from γGC and glycine. TF, transferrin; TfR1, transferrin-R1; LIP, labile iron pool; DMT1, divalent metal transporter 1; FPN-1, ferroportin-1; CP, ceruloplasmin; PUFA, polyunsaturated fatty acids; AA, arachidonic acid; AdA, adrenal acid; ACSL4, acyl-CoA synthase long chain family member 4; LPCAT3, lysophosphatidylcholine acyltransferase 3; ALOX15, Arachidonic acid-15-lipoxygenases; γGC, γ-Glutamic cysteine; GCL, Glutamic cysteine ligase; GSS, γ-Glutathione synthase. Template of cell membrane element come from SMART (https://smart.servier.com/smart_image/cell-membrane-15/).

Although iron ions dominate the process of ferroptosis, there is limited research on how lncRNAs regulate iron metabolism. The regulation of iron metabolism is mainly centered on the labile iron pool (LIP) capacity (*Richardson, 2004*). One of the main ways to regulate the capacity of intracellular iron pools is by controlling the input of iron. Transferrin-R1 (TfR1) is essential for intracellular iron absorption, and the abnormal accumulation of TfR1 on the cell surface serve as a specific markers of ferroptosis (*Mou et al., 2019*). Research has found that lncRNA PVT1 regulates the expression of TfR1 by sponging miR-214, thereby influencing the iron metabolism pathway (*Lu, Xu & Lu, 2020*). Moreover, the synthesis of various iron-containing proteins from unstable iron in the LIP is an important

pathway for utilizing intracellular iron ions. This process may be influenced by STEAP3. Studies have reported that in colon cancer, lncRNA STEAP3-AS1 can affect the expression of CDKN1C and impact the cell cycle by regulating STEAP3. Decreasing the expression of STEAP3-AS1 significantly inhibits tumor growth (*Na et al., 2020*). Given the function of STEAP3 in ferroptosis, it is possible that STEAP3-AS1 also affects iron ion metabolism in ferroptosis. It is yet to be confirmed whether reducing STEAP3-AS1 promotes the expression of STEAP3 to induce ferroptosis. Moreover, recycling iron ions by degrading intracellular iron-containing proteins is another way to increase the capacity of LIP in cells. URB1-AS1 co-localizes with ferritin in the cytoplasm, and their interaction encourages ferritin phase separation while preventing its lysosomal degradation and clearance (*Gao et al., 2023*). In bladder urothelial carcinoma, MAFG-AS1 promotes deubiquitination of PCBP2 by binding to poly(rC)-binding protein 2 (PCBP2), thereby stabilizing PCBP2. This protein interacts with FPN-1 to facilitate iron transport (*Xiang et al., 2021*). Therefore, transporting intracellular iron outside the cell is also a method to regulate and maintain cellular iron homeostasis.

So far, it appears that lncRNA primarily functions through decoy and guidance mechanisms, regulating iron metabolism at different stages of iron intake, utilization, storage, and release, thus affecting ferroptosis. In previous studies, PVT1 increases TfR1 expression by binding to it, leading to ferroptosis, whereas other lncRNAs have inhibitory effects (*Lu, Xu & Lu, 2020*). Different lncRNAs have varying effects on regulating ferroptosis due to their interactions with different target molecules. In addition, lncRNAs do not just interact with one or a few molecules during physiological processes. For example, PVT1 can also enhance the expression of tumor suppressor p53 by absorbing miR-214 (*Lu, Xu & Lu, 2020*). This multi-effector makes the regulation of biological activities by lncRNA more intricate and precise.

## LncRNA and lipid metabolism

Lipid peroxidation refers to the loss of hydrogen atoms in lipids under the action of free radicals or lipid peroxidation enzymes, leading to the oxidation, breakage, and shortening of lipid carbon chains (*Yin, Xu & Porter, 2011*). This process produces cytotoxic substances such as lipid free radicals, lipid hydroperoxides, and active aldehydes (malondialdehyde, 4-hydroxynonenal), resulting in lipid oxidative degradation reactions that cause cell damage (*Qiu et al., 2024*). Lipid peroxidation is harmful for the oxidative degradation of two important biofilm components, polyunsaturated fatty acids (PUFA) and phosphatidylethanolamine (PE), and poses a serious challenge to the maintenance of cell membrane integrity (*Pope & Dixon, 2023*). Lipid peroxidation free radicals can continuously participate in the oxidation process of PUFA, resulting in a cascade reaction characteristic of PUFA's lipid peroxidation reaction (Fig. 2B).

The research on lipid metabolism related to ferroptosis mainly focused on enzyme-catalyzed lipid peroxidation reactions. These enzymes are crucial targets for lncRNA regulation. Overexpression of ACSL4 leads to an increase in various PUFAs catalytic activity, altering the composition of cellular lipids and upregulating susceptibility to cell ferroptosis. This process is often monitored by lncRNAs through sponge action

(*Bouchaoui et al., 2023*). For example, lncRNA ZFAS1 regulates the expression of ACSL4 by competitively binding with miR-7-5p, leading to an increase in ACSL4 expression, and promoting ferroptosis in high glucose cultured human retinal endothelial cells (*Liu et al., 2022b*). NEAT1 influences ACSL4 by regulating miR-34a-5p and miR-204-5p, promoting docetaxel resistance in prostate cancer (*Jiang et al., 2020*). Knocking down H19 promotes the proliferation of cerebral microvascular endothelial cells and inhibits ferroptosis by modulating ACSL4 through sponging miR-106b-5p (*Chen et al., 2021a*). Furthermore, lncRNA TUG1 in exosomes produced by human urethral-derived stem cells can regulate ACSL4-mediated ferroptosis by interacting with SRSF1, alleviating renal ischemia/ reperfusion injury (*Sun et al., 2022*). The increased expression of PVT1 regulates the progression of atherosclerosis through miR-106b/5p/ACSL4 axis (*Zhang et al., 2023a*).

ALOX15 is activated during the catalytic process involving $Fe^{2+}$ (*Pope & Dixon, 2023*). The increase of ALOX15 expression led to an upregulation in lipid hydroperoxides PE-AA/AdA-OOH, which induces ferroptosis (*Ma et al., 2022*). By interfering with the expression of lncRNA ENST000000538705.1 in human coronary artery endothelial cells, the expression of ALOX15 is reduced, leading to alleviate myocardial injury in myocardial infarction rats (*Chen et al., 2022*). This reduction may also be related to the regulation of ferroptosis. There is limited literature on the regulation of ALOX15 by lncRNAs, and further research is needed.

Inhibiting the expression of lipid metabolism-related enzymes reduces the accumulation of intracellular lipid peroxides, which helps prevent ferroptosis. It seems that the regulation of enzymes is mostly mediated by the decoy action of lncRNA, triggering ferroptosis through the ceRNA mechanism. LncRNA TUG1 inhibits ferroptosis by interacting with the splicing regulatory factor SR protein family, which the affects the stability of ACSL4 mRNA (*Sun et al., 2022*). This is consistent with TUG1's targeting miR-494-3p/E-cadherin in AKI caused by ischemia/reperfusion, where TUG1 overexpression significantly reduces kidney injury and cell apoptosis, indicating that the same molecule can produce the same or similar effects *via* different mechanisms (*Chen, Xu & Tan, 2021*). Although there is not much research on lncRNAs affecting lipid metabolism, further studies in this area will reveal more effectors involved as our understanding of lipid metabolism deepens.

## LncRNA and glutathione metabolism

Glutathione is a water-soluble tripeptide composed of cysteine, glutamic acid, and glycine residues. There are two forms of glutathione in the body, one is reducing glutathione (GSH), and the other is oxidized glutathione (GSSG) (*Lapenna, 2023*). GSH is a major antioxidant, reducing $H_2O_2$ to $H_2O$, balancing intracellular free radical levels, and serving as a cofactor for glutathione peroxidase 4 (GPX4) in reducing lipid hydroperoxide LOOH. GSH repairs LOOH in the biofilm, and helps prevent ferroptosis (*Lapenna, 2023*). However, when the activity of the System Xc- subunit is inhibited, leading to insufficient cystine intake, the synthesis of glutathione is blocked, resulting in inhibited GPX4 activity, and decreased the ability to resist lipid peroxidation (Fig. 2C) (*Ye et al., 2022*).

Compared with regulating iron and lipid metabolism, the research on antioxidant metabolism associated with ferroptosis is more thorough. These studies typically focus on two main aspects. Firstly, lncRNA primarily regulates miRNA through the ceRNA mechanism, causing changes in the expression of miRNA target molecules and interfering with ferroptosis-related pathways. For instance, lncRNA OIP5-AS1 has been reported to regulate SCL7A11 through miR-128-3p, inhibiting ferroptosis in prostate cancer cells induced by long-term exposure to cadmium (*Zhang et al., 2021*). Secondly, lncRNA can indirectly or directly modify key proteins, affect protein expression, and thereby regulate ferroptosis. Recent research indicates that lncRNA DUXAP8 enhances the effect of SLC7A11 and inhibits cell ferroptosis by promoting palmitoylation of SLC7A11 and preventing lysosomal degradation of the protein (*Shi et al., 2023*). In the antioxidant process, SLC7A11-AS1 has an important role, as it can target SLC7A11 to down-regulate the expression and inhibit the invasion of ovarian cancer epithelial cells. In testicular-derived cancer cells, SLC7A11-AS1 represses the expression of SLC7A11, thus reducing GSH levels, increasing ROS levels, and enhancing lipid peroxidation. These observations promote cell ferroptosis, and are closely related to male infertility (*Sanei-Ataabadi, Mowla & Nasr-Esfahani, 2020*). Although reports on SLC7A11 are common, little is known about the regulatory mechanism of SLC3A2. LncRNA SNHG1 activates the Akt pathway by regulating SLC3A2, which contributes to sorafenib resistance (*Li et al., 2019*).

Regulating the anti-lipid peroxidation enzyme GPX4 is also a key strategy for regulating ferroptosis. The mechanism by which lncRNA regulates GPX4 is sponge action. For example, lncRNA PVT1 prevents ferroptosis in liver cancer cells by targeting miR-214-3p, which stops the latter from binding to the 3′-UTR of GPX4 mRNA. In addition, RUNX1-IT1 is believed to be able to promote the formation of IGF2BP1 liquid-liquid phase separation (LLPS) biomolecular condensates by directly binding to the N6-methyladenosine m6A reader IGF2BP1. This results in IGF2BP1 occupying more GPX4 mRNA, improving the stability of GPX4 mRNA, promoting the expression of GPX4, and inhibiting breast cancer cell death (*Wang et al., 2023a*). The regulation of GPX4 by lncRNA also occurs through translation, as seen with lnc-HCP5 (Fig. 1F) (*Tong et al., 2023a*).

Compared to normal tissues, the expression levels of SCL7A11 and GPX4 are significantly increased in tumor tissues, including colon adenocarcinoma (*Cheng et al., 2022*), renal clear cell carcinoma (*Zou et al., 2019*), prostate cancer (*Ghoochani et al., 2021*), thyroid cancer (*Sekhar et al., 2022*), and endometrial cancer (*Chen et al., 2024*). The identified lncRNAs mostly enhance the expression of both and inhibit ferroptosis. The expression of SNHG15 is increased in GC. It can sponge miR-24-3p to protect E2F1 and MYC from the degradation of miR-24-3p. On the one hand, E2F1 and MYC can activate the expression of SNHG15, forming positive feedback and promoting the occurrence of gastric cancer. On the other hand, SNHG15 competitively binds to hnRNPA1 to prevent its mRNA degradation, stabilize SLC7A11 mRNA, and inhibit ferroptosis (*Duan et al., 2024*). It is suggested that SCL7A11 and GPX4, may be oncogenes. Cancer cells often prevent ferroptosis and promote cell survival by upregulating the expression of these

**Table 1  LncRNAs involve in the regulation of SLC7A11 and GPX4.**

| Molecules | LncRNAs | Mechanisms | Cell resources | Effect on ferroptosis | Reference |
|---|---|---|---|---|---|
| SLC7A11 | BBOX1-AS1 | Regulating SCL7A11 expression through sponge action of miR-513a-3p. | Esophageal squamous cell cancer | Contributory | *Pan et al. (2022)* |
| | SLC7A11-AS1 | Directly targeting SLC7A11 to downregulate the expression. | Colorectal cancer | Contributory | *Wang et al. (2023b)* |
| | SLC16A1-AS1 | Promoting the expression of SLC7A11 *via* the inhibition of miR-143-3p by ceRNA mechanism. | Renal cell carcinoma | Contributory | *Li et al. (2022)* |
| | HEPFAL | Promoting the ubiquitination of SLC7A11 and reduce its stability. | Hepatocellular carcinoma | Contributory | *Zhang et al. (2022a)* |
| | PMAN | Increasing the stability of SLC7A11 mRNA by promoting the distribution of ELAVL1 in the cytoplasm. | Gastric cancer | Contributory | *Lin et al. (2022)* |
| | NEAT1 | Competitively binding miR-150-5p, which directly binds to BAP1 to regulate the expression of SLC7A11. | SK-N-SH cells | Contributory | *Zhao et al. (2022)* |
| | SNHG14 | Targeting miR-206 to regulate the expression of SLC7A11 *via* sponge action. | Osteosarcoma | Inhibitory | *Li et al. (2023c)* |
| | LINC00578 | Binding ubiquitin ligase E2K to reduce ubiquitination of SLC7A11 | Pancreatic cancer | Inhibitory | *Li et al. (2023a)* |
| | LINC00618 | Reducing the expression of lymphoid specific hepatitis (LSH), which can accumulate to the SLC7A11 promoter region and enhance the transcription of SLC7A11. | Leukemia cells | Inhibitory | *Wang et al. (2021a)* |
| | DUXAP8 | Promoting palmitoylation of SLC7A11 and preventing lysosomal degradation of the protein. | Hepatocellular carcinoma | Inhibitory | *Shi et al. (2023)* |
| | FLVCR1-AS1 | Inhibiting the role of miR-23a-5p as ceRNA to promote the expression of SLC7A11. | Cervical cancer | Inhibitory | *Zhou et al. (2022)* |
| | OIP-AS1 | Regulating SCL7A11 through sponge action of miR-128-3p. | Prostate cancer | Inhibitory | *Zhang et al. (2021)* |
| | Uc.339 | Competitively binding to pri-miR-339 to inhibit the production of mature miR-339 and promoting SLC7A11 expression. | Lung adenocarcinoma | Inhibitory | *Zhang et al. (2022b)* |
| | PCAT1 | Improving the stability of c-Myc protein and promoting the transcription of SLC7A11 | Prostatic cancer | Inhibitory | *Jiang et al. (2022)* |
| | DLEU1 | Promoting the degradation of ATF3 mRNA by binding with ZFP36, thereby upregulating the expression of SLC7A11 | Glioblastoma | Inhibitory | *Zhao et al. (2023)* |
| | AGAP2-AS1 | Increasing mRNA stability of SLC7A11 through the IGF2BP2 pathway | Melanoma | Inhibitory | *An et al. (2022)* |
| GPX4 | OTUD6B-AS1 | Binding to HuR to stabilize TRIM16 mRNA, which promotes the ferroptosis mediated by GPX4 | Colorectal cancer | Contributory | *Zhang et al. (2023b)* |
| | MEG3 | Reducing the binding of p53 to GPX4 promoter by directly targeting p53, thereby promoting ferroptosis | RBMVECs | Contributory | *Chen et al. (2021b)* |
| | PVT1 | Directly interacting with miR-214-3p to suppress the binding of miR-214-3p and 3′ UTR of GPX mRNA | Liver cancer | Inhibitory | *He et al. (2021)* |
| | linc00976 | Regulate GPX4 expression by competitive binding with miR-3202 | Cholangiocarcinoma | Inhibitory | *Lei et al. (2022)* |
| | LINC01134 | Promoting Nrf2 recruitment to the GPX4 promoter region to exert transcriptional regulation of GPX4 | Hepatocarcinoma | Inhibitory | *Kang et al. (2022)* |

| | | | | | |
|---|---|---|---|---|---|
| **Table 1** (continued) | | | | | |
| Molecules | LncRNAs | Mechanisms | Cell resources | Effect on ferroptosis | Reference |
| | ADAMTS9-AS1 | Inhibiting GPX4 expression *via* the ceRNA action of miR-6516-6p | Endometrial stromal cells | Inhibitory | *Wan et al. (2022)* |
| | HCG18 | Adsorbing miR-450b-5p as a sponge to promotes GPX4 expression | Hepatocellular carcinoma | Inhibitory | *Li et al. (2023d)* |
| | RUNX-IT1 | Directly binding to N6-methyladenosine m6A reader IGF2BP1 and promoting the formation of liquid-liquid phase separated biomolecular condensates, improving the stability of GPX4 mRNA | Breast cancer | Inhibitory | *Wang et al. (2023a)* |

molecules. This suggests that tumors can be treated by developing drugs targeting these targets or corresponding lncRNA molecules (see later). The lncRNAs involved in GPX4 and SLC7A11 regulation and their mechanisms are summarized in Table 1.

# LNCRNA REGULATES THE FERROPTOSIS RELATED PROCESSES

Multiple signaling pathways can regulate ferroptosis by influencing iron metabolism, lipid metabolism, and antioxidant metabolism (*Hassannia, Vandenabeele & Vanden Berghe, 2019*). At present, the signal pathways reported in the literature mainly focus on Nrf2, autophagy, and p53, while other pathways such as Notch and the cell cycle are occasionally reported. Understanding the regulatory mechanisms of these pathways contributes to a comprehensive understanding of how ferroptosis affects cell fate. A review of related pathways is as follows.

## Nrf2 pathway

One of the characteristics of ferroptosis is the accumulation of a large amount of intracellular lipid ROS when the cell's antioxidant capacity decreases (*Pope & Dixon, 2023*). The antioxidant system plays a crucial role in ferroptosis. As a key transcription factor regulating antioxidant stress, downstream genes of n Nuclear Factor erythroid 2-Related Factor 2 (Nrf2/NEF2L2) include NADPH, HO-1, SLC7A11/xCT, NADPH quinone oxidoreductase 1 (NQO1), thioredoxin 1, phase II detoxifying enzymes (such as GSH S-transferase, UDP glucuronosyltransferase, GPX4, GSH reductase, and GCLc and GCLm) and so on (*Osama et al., 2020*). Due to the crucial role of Nrf2 in antioxidant activity, it has regulatory effects on many key molecules involved in ferroptosis.

LncRNAs can regulate multiple levels, including Nrf2 expression (Fig. 3). The competitive binding of lncRNA SNAI3-AS1 to SND1 disrupts the m6A-dependent recognition of 3′-UTR in Nrf2 mRNA by SND1, ultimately reducing the mRNA stability of Nrf2 and promoting ferroptosis in gliomas (*Zheng et al., 2023a*). Additionally, there are reports indicating that Nrf2 can directly or indirectly regulate the expression and function of crucial molecules in ferroptosis. Nrf2 can directly bind to the promoter of the SLC7A11

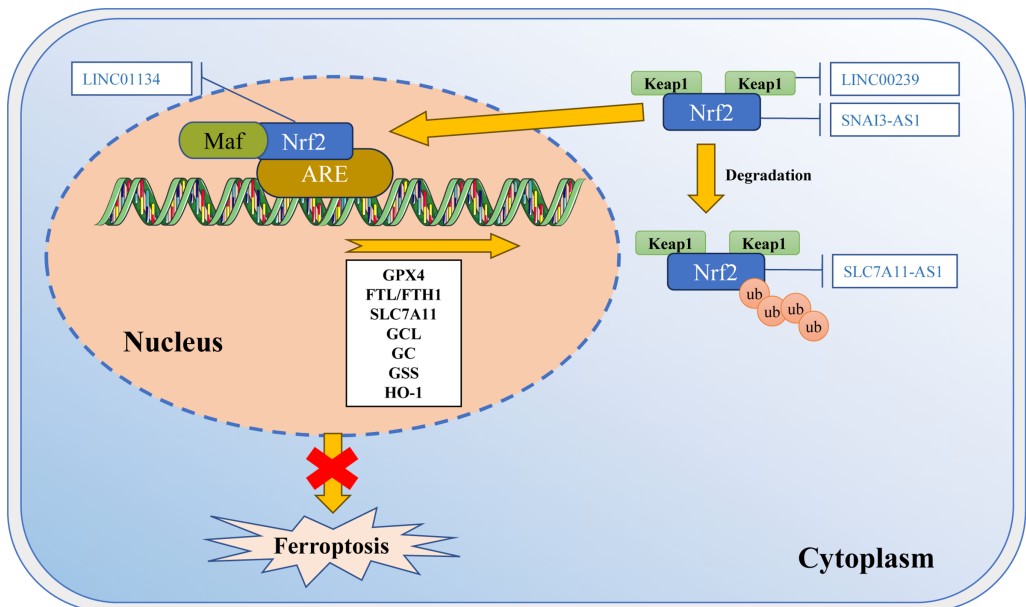

**Figure 3 Regulation of Nrf2 transcription machinery through lncRNAs involves in ferroptosis.** The activity of Nrf2 is strictly regulated by Keap1. Keap1 passively separates Nrf2 from the cytoplasm, and plays a positive role in targeting Nrf2 for ubiquitination and proteasome degradation. Under normal circumstances, Nrf2 binds to Keap1 and is inactivated due to ubiquitination and degradation in proteasomes. Once the cell is in an oxidative stress state, Nrf2 will be released from the Keap1 binding site and rapidly translocated into the nucleus. It then interacts with the antioxidant response element (ARE) in the target gene promoter, activating the transcription pathway to balance oxidative stress and maintain the cell's redox homeostasis. SNAI3-AS1 reduces the mRNA stability of Nrf2. LINC00239 interacts with Keap1 to disrupt the Keap1/Nrf2 complex. SLC7A11-AS1 blocks the degradation of Nrf2. LINC01134 promotes Nrf2 to the GPX4 promoter region, enhances GPX4 expression. Template of nucleic acid element come from SMART (https://smart.servier.com/smart_image/dna/).

subunit to promote the expression of SLC7A11 (*Habib et al., 2015*). It also controls the catalysis of GPX4, GCL, glutathione reductase, and GSS, as well as the regulation of their subunits. For example, LINC01134 can promote Nrf2 to the GPX4 promoter region, enhance GPX4 expression, and inhibit ferroptosis in HCC cells (*Kang et al., 2022*). Furthermore, lncRNA can impact the degradation of Nrf2. The lncRNA SLC7A11-AS1 can promote the chemoresistance in pancreatic cancer cells by blocking the degradation of Nrf2 mediated by SCF (*Yang et al., 2020*). Moreover, it has been observed that LINC00239 promotes the proliferation of colorectal cancer cells by interacting with Keap1, leading to instability of the Keap1/Nrf2 complex. By inhibiting the ubiquitination of Nrf2 protein, it enhances the stability of Nrf2 protein, promotes CRC invasion, and promotes LINC00239 transcription in a positive feedback manner (*Han et al., 2022*).

According to recent studies, ferroptosis inducers RSL3 and ML-162 activate the endoplasmic reticulum (ER)-related pathways through the PERK-ATF4-SESN2 axis. This leads to an increase in p62 expression (*Shin et al., 2018*). P62 then inactivates Keap1, while Nrf2 is activated through the interaction between p62-Keap1. As a result, Nrf2 promotes the association with iron and antioxidant system ARE, which leads to a decrease in LIP

(*Shin et al., 2018*). These observations suggest that Nrf2 may play a role in regulating the process of ferroptosis by cross-linking with autophagy and ER stress.

## Autophagy

Autophagy is the cellular process by which cells remove and degrade unused proteins and cellular components, as well as damaged organelles, through lysosomes. Depending on the different substrates degraded during autophagy, autophagy can be categorized into mitophagy, pexophagy, lipophagy, ferritinophagy, and so on (*Levine & Kroemer, 2019*). Previous reports have suggested that ferritinophagy may contribute to ferroptosis.

Ferritinophagy is a selective degradation process targeting ferritin. Previous studies have shown that increased ferritin levels can inhibit ferroptosis, while ferritinophagy can promote ferroptosis by degrading ferritin and releasing free iron ions from the ferritin heavy chain 1 (FTH1) (*Jin et al., 2023a*). LncRNA CACNA1G-AS1 has been found to upregulate the expression of FTH1 through the IGF1BP2 axis, regulate ferritinophagy to suppress ferroptosis, and ultimately promoting the proliferation and migration of ovarian cancer cells (*Jin et al., 2023b*). Despite ferritin's crucial role in iron homeostasis regulation, there are limited reports on the adjustment of ferroptosis by lncRNA *via* ferritin.

Interestingly, mitophagy and lipophagy may also be cross-linked with ferroptosis. Mitophagy seems to have a bidirectional regulatory effect on ferroptosis. In the early stages of iron overload, mitophagy can protect cells by preventing the release of ROS from dysfunctional mitochondria (*Li et al., 2023b*). However, excessive mitophagy might lead to increased iron levels, which can amplify lipid peroxidation and ultimately trigger ferroptosis (*Rademaker et al., 2022*; *Yang et al., 2023a*). Research has reported that overexpression of MEG3 inhibits the expression of Rac1 by interacting with Rac1's 3′UTR, reducing the generation of ROS. MEG3 can prevent co-localization of Rac1 and FUNDC1, enhancing FUNDC1-mediated mitophagy (*Bi et al., 2024*; *Wang et al., 2021b*). Additionally, overexpression of TMEM161B-AS1 downregulates the expression of FANCD2 by playing a sponge role in regulating miR-27a-3p, and regulating ferroptosis in glioma cells. FANCD2 is a key repair factor in DNA repairing, and takes part in the mitophagy *via* the interaction with Parkin to clear the damaged mitochondria (*Sumpter et al., 2016*). Although the role of lncRNA in mitophagy has gained significant attention, the mechanism of its cross-linking effect in ferroptosis has not yet been fully elucidated.

Lipophagy is a unique selective autophagy that targets lipid droplets and regulates cellular lipid levels. Lipophagy promotes ferroptosis by reducing lipid storage and promoting lipid peroxidation (*Lan et al., 2023*; *Liang, Minikes & Jiang, 2022*). Knocking down the lipid drop cargo receptor RAB7A can prevent TSL3-induced lipid peroxidation and subsequent ferroptosis (*Bai et al., 2019*). There are literature reports that upregulated TMEM147-AS1 competitively binds to has-let-7 through the ceRNA mechanism, leading to continuous phosphorylation of GSK3β and nuclear translocation of β-catenin (*Shao et al., 2023*). Although it has not been studied whether TMEM147-AS1 regulates the process of ferroptosis, GSK3β/β-catenin has been proven to regulate ferroptosis (*Liu et al., 2022a*). These studies indicate that the balance between lipid storage and degradation is

closely related to the cell's response to ferroptosis, and targeting the lipophagy pathway may be one of the new strategies for inhibiting ferroptosis.

### p53 pathway

As an important tumor suppressor protein, p53 exerts tumor-suppressive function by selectively regulating many targeted genes to adjust basic cellular responses (*Liu et al., 2024*). In normal cells, the expression of p53 protein is usually maintained at a low level. Once activated, p53 binds to the p53-responsive element in the target gene to transcriptionally regulate gene expression, thereby affecting various cellular responses, including apoptosis, DNA repair, antioxidant defense, cell cycle arrest, autophagy, and ferroptosis (*Jiang et al., 2015*; *Liu et al., 2024*). According to reports, p53 binds to the promoter region of the RNA binding protein ELAVL1 (ELAV type RNA binding protein 1) to inhibit the expression of ELAVL1. ELAVL1 can bind and stabilize LINC00336. In lung cancer cells, LINC00336 acts as a sponge for miR-6852, thereby enhancing cystathionine β synthetase (CBS) activity, the marker of ferroptosis to sulfur pathway, to repress ferroptosis (*Wang et al., 2019a*). In terms of lipid metabolism and antioxidant metabolism, PELATON (LINC01272) can form a complex by binding the RNA binding protein EIF4A3 to p53, and act as a ferroptosis inhibitor through mutant p53 mediated ROS ferroptosis pathway. This inhibition involves reducing ROS production, reducing the level of divalent iron ions, promoting the expression of SLC7A11, and suppressing the expression of ACSL4 and Cox2 (*Fu et al., 2022*). The first and 871st nucleotides in lncRNA P53RRA bind to the RMM interaction region of Ras GTPase-activating protein-binding protein 1 (G3BP1) to form a complex. This complex replaces p53 in G3BP1, preventing G3BP1 from intercepting p53 in the cytoplasm. Consequently, this promotes more p53 to enter the nucleus and activating the p53 signaling pathway, possibly inducing ferroptosis by affecting the transcription of metabolic-related genes SLC1A5, SLC7A11, SLC2A4, *etc*., (*Mao et al., 2018*). These research confirm that p53 can induce ferroptosis by regulating lncRNA. It is also influenced by many ncRNAs, participating in the regulation of iron metabolism, lipid metabolism, and antioxidant metabolism processes.

### Others processes

Ferroptosis is often closely related to tumors. These key molecules involve physiological processes such as the Notch pathway and cell cycle. However, the exact mechanism is still unclear. The most common is that lncRNAs act as decoys, affecting miRNA or circRNA through ceRNA effects. This, in turn, affects the expression of key molecules related to each pathway and regulates cellular biological processes. Inhibiting lncRNA MEG8 can upregulate miR-497-5p, thus inhibiting the expression of Notch2 and inducing ferroptosis in vascular endothelial cells (*Ma et al., 2021a*). In high glucose (HG) treated cardiomyocytes, ZFAS1 regulates the expression of miR-150-5p. MiR-150-5p can salvage the expression of CCND2, while its inhibition reduces mitochondrial membrane potential and CCND2 expression, indicating that ZFAS1 can restrain ferroptosis by targeting miR-150-5p (*Ni et al., 2021*). AC005332.7 can regulate CCND2 and affect cell cycle by binding to miR-331-3p, thereby inhibiting ferroptosis in AC16 cells under oxygen and glucose

deprivation conditions (*Dai et al., 2023*). LncRNA exerts a decoy effect through the ceRNA mechanism to regulate the expression of key molecules, which seems to be the main way in which lncRNA regulates ferroptosis and related physiological processes. The rapid development of scientific research has enriched and deepened the understanding of ferroptosis and lncRNAs. Due to space constraints, we are unable to cover all the research and have only sorted and reviewed the main information. Links between ferroptosis and other cellular physiological activities have fully confirmed the crucial role of ferroptosis in cell fate. The elucidation of the crosstalk between ferroptosis and these pathways can help us further understand programmed cell death.

## PROSPECT

Ferroptosis is closely related to the occurrence and development of many diseases, and can be triggered by different pathological conditions (*Doll et al., 2017*; *Wang et al., 2019b*). Many tumor suppressors have been proven to make cells sensitive to ferroptosis, such as p53, tumor suppressor BRCA1-related protein 1 (BAP1) (*Zhang et al., 2018b*), KEAP1 (*Han et al., 2022*), *etc*. Ferroptosis has been identified as one of the important causes of tumor cell death, such as non-small cell lung cancer (*Wang et al., 2024b*), breast cancer (*Tong et al., 2023a*), and pancreatic cancer (*Rademaker et al., 2022*). However, ferroptosis may also play a role in development of cancer in the context of tumor immunity. For example, CD36 mediates the uptake of fatty acids by tumor infiltrating CD8+ T cells in the tumor microenvironment, while increased expression of CD36 can induce lipid peroxidation and ferroptosis in CD8+ T cells, thereby inhibiting anti-tumor immunity (*Han et al., 2024*; *Ma et al., 2021b*). In this case, inhibiting CD36 expression or preventing ferroptosis of CD8+ T cells can effectively restore anti-tumor activity. Nonetheless, some lncRNAs have been shown to have the potential as disease biomarkers and therapeutic targets due to their stability and accessibility, without the limitation of invasive acquisition methods (*Jin, Huang & Tian, 2024*; *Lekva et al., 2024*; *Trevisani et al., 2022*). Targeting lncRNA may serve as one of the tumor inhibition mechanisms by which cells exert tumor suppressive effects to clear malignant transformed cells, creating new opportunities for tumor diagnosis, treatment, and intervention.

Exploring the different mechanisms by which lncRNA regulates ferroptosis can help develop promising new therapeutic strategies. Unlike mRNA and miRNA, the primary sequence of lncRNA exhibits little conservation (*Kutter et al., 2012*). LncRNA binds to target genes through complementary base pairing to directly regulate the transcriptional translation of target genes or indirectly regulate the transcriptional translation of upstream or downstream genes of target genes (such as decoy function) (*Majumdar et al., 2024*; *Na et al., 2020*). This means that although many studies have shown that lncRNA functions by a mechanism similar to ceRNA *via* primary structure, the low sequence conservation poses a huge challenge for the targeting of these lncRNAs.

The difference is that the secondary and tertiary structures of lncRNA are highly conserved (*Nitsche & Stadler, 2017*). The secondary structure of lncRNA can regulate the state of chromatin, thereby affecting gene expression. For example, steroid receptor RNA activator lncRNA (lncRNA-SRA) is a kind of lncRNA related to breast cancer, which can

**Table 2 Mechanisms and structures of lncRNAs in the regulation of key molecules in ferroptosis.**

| Processes | Molecules | lncRNAs | Actions | Mechanisms | Reference |
|---|---|---|---|---|---|
| Iron metabolism | TfR1 | PVT1 | Primary | PVT1 exerts sponge effect to adsorb miR-214 and upregulates TFR1 expression. | *Lu, Xu & Lu (2020)* |
| | STEAP3 | STEAP3-AS1 (Undefined) | Primary | Regulating its adjacent STEAP3 to affect the expression of CDKN1C in *cis*, thereby influencing the cell cycle. | *Na et al. (2020)* |
| | FPN-1 | MAFG-AS1 | Secondary | Recruiting UCHL5 and binding to poly(rC)-binding protein 2 (PCBP2) to deubiquitinate PCBP2, and promote iron transport. | *Xiang et al. (2021)* |
| | FTH1 | URB1-AS1 | Secondary | Binding with ferritin, promoting liquid-liquid phase separation, and reducing free iron ions. | *Gao et al. (2023)* |
| | | LINC00152 | Primary | Increasing mRNA stability of PDE4D by binding to PDE4D 3′-UTR, and preventing ferroptosis | *Saatci et al. (2024)* |
| Lipid metabolism | ALOX15 | ENST00000538705.1 | Undefined | Undefined | *Chen et al. (2022)* |
| | ACSL4 | ZFAS1 | Primary | Competitive binding with miR-7-5p to regulate ACSL4 expression. | *Liu et al. (2022b)* |
| | | NEAT1 | Primary | Competitive binding with miR-34a-5p and miR-204-5p to regulate ACSL4 expression | *Jiang et al. (2020)* |
| | | PVT1 | Primary | Competitive binding with miR-106b-5p to regulate ACSL4 expression | *Zhang et al. (2023a)* |
| | | TUG1 | Secondary | Regulating the stability of ACSL4 mRNA by interacting with RNA binding protein SRSF1 | *Sun et al. (2022)* |
| Anti-oxidation metabolism | GPX4 | TMEM44-AS1 | Secondary | Binding to IGF2BP2 enhances the stability of GPX4 mRNA. | *Yang et al. (2023b)* |
| | | MEG3 | Secondary | Binding to PTBP1 to degrade GPX4. | *Zhang et al. (2024)* |
| | SLC7A11 | SNHG15 | Secondary | Competitive binding to HNRNPA1 to prevent SLC7A11 mRNA degradation. | *Duan et al. (2024)* |
| | | LncRNA-casc2 | Secondary | Targeting FMR1 and increasing the stability of SOCS2 mRNA to promote ubiquitination degradation of SLC7A11. | *Wang et al. (2024c)* |
| | | ROR1-AS1 | Secondary | Enhancing the stability of SLC7A11 mRNA through interaction with IGF2BP1. | *Yao et al. (2024)* |
| Nrf2 pathway | Nrf2 | PSMA3-AS1 | Primary | Competitively binding to miR-224-3p downregulates Nrf2 expression. | *Qiu et al. (2023)* |
| | | SNAI3-AS1 | Secondary | Competitively binding to SND1 to interfere with the m6A dependent recognition of 3′-UTR in Nrf2 mRNA by SND1. | *Zheng et al. (2023a)* |
| | | LINC01134 | Secondary | Promoting the recruitment of Nrf2 to the GPX4 promoter. | *Kang et al. (2022)* |
| | | SLC7A11-AS1 | Secondary | Interacting with β-TRCP1, and blocking SCFβ-TRCP-mediated ubiquitination, thereby degrading Nrf2 protein. | *Yang et al. (2020)* |
| | | LINC00239 | Secondary | Interacting with Keap1 to enhance the stability of Nrf2 protein by inhibiting its ubiquitination. | *Han et al. (2022)* |
| Ferritinophagy | NCOA4 | Lnc-HZ06 | Secondary | Promoting SUMOylation of HIF-1α by inhibiting SENP1-mediated deSUMOylation, and further inducing ferroptosis *via* NCOA4. | *Tian et al. (2024)* |
| | Angptl4 | Mir22hg | Secondary | Enhancing the stability of Angptl4 mRNA by recruiting YTHDC1. | *Deng et al. (2024)* |
| | FTH1 | CACNA1G-AS1 | Secondary | Upregulating FTH1 expression through IGF1BP2 axis, regulating ferritin autophagy. | *Jin et al. (2023b)* |
| Mitophagy | Rac1 | MEG3 | Primary | Interacting with 3′-UTR of Rac1 mRNA to inhibit the expression of Rac1, thereby enhancing FUNDC1-mediated mitophagy. | *Wang et al. (2021b)* |

| Table 2 (continued) | | | | | |
|---|---|---|---|---|---|
| Processes | Molecules | lncRNAs | Actions | Mechanisms | Reference |
| p53 pathway | p53 | LINC00336 | Primary | Acting as a sponge for miR-6852 to enhance the activity of cysteine β-synthase (CBS). | *Wang et al. (2019a)* |
| | | PELATON | Secondary | Binding with EIF4A3 and p53 to inhibit ferroptosis. | *Fu et al. (2022)* |
| | | ITGB2-AS1 | Secondary | Inhibition of p53 expression by increasing NAMPT expression through binding with FOSL2. | *Chen et al. (2023)* |
| Notch pathway | Notch2 | MEG8 | Primary | Sponging with miR-497-5p expression to regulate NOTCH2 expression. | *Ma et al. (2021a)* |
| | FANCD2 | TMEM161B-AS1 | Primary | Downregulating the expression of FANCD2 and CD44 by exerting sponge effects to regulate miR-27a-3p. | *Chen et al. (2021c)* |
| Cell cycle | CCND2 | ZFAS1 | Primary | sponge with miR-150-5pto regulate the expression of CCND2. | *Ni et al. (2021)* |

activate several sex hormone receptors and has highly conserved helix, terminal ring and bulge in many species (*Novikova, Hennelly & Sanbonmatsu, 2012*). Another example is HOTAIR, which functions as a scaffold through its secondary structure (see above) (*Tsai et al., 2010*). LncRNA can also form special tertiary structures to perform functions. For instance, the lncRNA maternal expression gene 3 (MEG3) contains three predicted structural motifs that are conserved across its multiple subtypes (*Zhang et al., 2010*). Two of these three motifs are necessary for activating the tumor suppressor p53, while the other is involved in inhibiting DNA synthesis (*Uroda et al., 2019*; *Zhu et al., 2015*). *Chillon & Marcia (2020)* reviewed the relevant advances in the structure and function of lncRNA. The secondary and tertiary structures of lncRNA can reveal information about possible interaction partners and transcriptional functions.

In the process of ferroptosis, we summarized the regulatory molecules and mechanisms of key pathways (Table 2), such as SLC7A11, TFR1, *etc.*, in iron ion metabolism. The mechanism action of most lncRNAs is exerted through the primary structure (by sponge effect). Some lncRNAs can also function in a secondary structure through signal or scaffold. In fact, small molecules that bind to RNA targets have been developed for use as bioactive degradation agents, selectively downregulating pathogenic RNA (*Tong et al., 2023b*). However, the regulation of ncRNAs by key molecules of ferroptosis seems to be diverse, and there may be compensatory mechanisms among them. In addition, the high-resolution structure of lncRNAs is still in its early stages, and the molecular level details of the mechanism of lncRNAs are not yet clear, so their structure function relationship has not been determined (*Spokoini-Stern et al., 2020*; *Uroda et al., 2020*). There are still many issues in the process of targeting lncRNA as a tumor suppressive mechanism for clearing malignant transformed cells. On the one hand, although RNA structure detection methods *in vitro* have developed rapidly, the lncRNA *in vivo* is usually different from that *in vitro* with stronger dynamics (*Chen, 2022*; *Martens et al., 2021*). On the other hand, the ability to penetrate cells on targeted molecules severely challenges their application *in vivo*. Although we have continuously enriched our research on ferroptosis

and its regulatory mechanisms, utilizing it as a weapon still requires addressing the structure of lncRNA, which still has a long way to go.

### Funding

This work was supported by the Natural Science Foundation of Hunan Province (No. 2022JJ40367), the Research Foundation of Education Bureau of Hunan Province (21B0435), and the Hunan Province College Students' Research, Learning, and Innovative Experiment Project (No. S202210555308, and X202310555115). The funders had no role in study design, data collection and analysis, decision to publish, or preparation of the manuscript.

### Grant Disclosures

The following grant information was disclosed by the authors:
Natural Science Foundation of Hunan Province: 2022JJ40367.
Research Foundation of Education Bureau of Hunan Province: 21B0435.
Hunan Province College: S202210555308 and X202310555115.

### Competing Interests

The authors declare that they have no competing interests.

### Author Contributions

- Yating Wen conceived and designed the experiments, analyzed the data, authored or reviewed drafts of the article, and approved the final draft.
- Wenbo Lei analyzed the data, authored or reviewed drafts of the article, and approved the final draft.
- Jie Zhang performed the experiments, prepared figures and/or tables, and approved the final draft.
- Qiong Liu performed the experiments, prepared figures and/or tables, and approved the final draft.
- Zhongyu Li conceived and designed the experiments, authored or reviewed drafts of the article, and approved the final draft.

### Data Availability

This review article does not include experimental data or code.

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
