# Peer review of "Advances in understanding the role of lncRNA in ferroptosis"

_PeerJ, doi:10.7717/peerj.17933_

## Round 0.1 · original submission · Major Revisions

As you can see, both reviewers raised several serious concerns and requested revision. Please address all the issues pointed by the reviewers and amend manuscript accordingly.

Reviewer 1 ·

Basic reporting

The review article titled ‘Advances in Understanding the Role of lncRNA in Ferroptosis’ summarized the current knowledge about the role of lncRNA in regulating ferroptosis involved in multiple diseases and biological processes. It has potential to benefit readers studying both lncRNA and ferroptosis.

This field has been reviewed in 2022 by Amar et al., which focuses on ferroptosis and cancer. There are overlaps between the two reviews. However, the current manuscript reviews the role of lncRNA in a broader scope than just cancer.

Experimental design

While this review holds the potential to benefit researchers, it has several major issues that must be addressed.

A major issue is the accuracy. See the following examples:
1) In lines 110-111, the sentence is incorrect. There are lncRNAs transcribed from the sense-strand of protein-coding genes. They can be in the introns of the protein-coding mRNAs or in the UTRs. lncRNA DAPALR is an example.
2) In lines 127-129, the explanation of cis and trans is incorrect. Cis means on the same mRNA/gene. Also, there is no reference to support the claims in lines 124-132.
These are just examples. The whole manuscript should be checked to remove misleading sentences.

A second major issue is that some parts of the manuscript are confusing. See the following examples:
1) In lines 52-53, the authors claimed that ferroptosis involves the development of cancer. However, in lines 71-72, it is said that ferroptosis is anti-cancer. How can something involve cancer development as well as be anti-cancer?
2) In lines 125-126, how can transcription factors and RBPs locate on DNA sequences? Transcription factors are more like proteins binding to DNA while RBPs stand for RNA-binding proteins.

A third major issue is the lack of references. The authors only include references for the example studies reviewed in the article, with little or no references for the background knowledge. For example, lines 36-37, 42-44, 76-79, 81-82, 159-160 all need references to support. The lack of references for such statements is extremely common throughout the manuscript, which is uncommon for scientific articles.

Besides, the choice of references is not appropriate. For example, in lines 52-53, the reference (Chen et al., 2021c) for cancer and ischemia is actually a review article, not the original research paper. Also, in Chen et al., 2021c, ischemia is only mentioned in the abstract as ‘tissue ischemia-reperfusion injury’. How can this support the linkage between ferroptosis and ischemia?

Also, most of the claims, if with reference, are only supported by one reference and many of the claims share one reference. A review article should have adequate references to ensure the points included are solid. To my understanding, a single reference is not always enough.

Validity of the findings

There is a prospect section talking about future studies, but I would suggest expanding it.

Additional comments

This review article should be carefully revised before it can be considered for publication to avoid misleading information.

Reviewer 2 ·

Basic reporting

The manuscript by Zhongyu et.al discussed about the role of lncRNAs in ferroptosis. The topic is timely and would be interesting to research community. However, the manuscript needs more changes before consideration. There are many key references related to lncRNAs and their biological function that are missing in the manuscript. I am providing my main suggestions.

Major concerns

One of the major concerns is that the authors just provided a literature review linking the role of lncRNAs and ferroptosis from the published research. There is no perspective from the author's point of view, For instance, the prospect paragraph should be well elaborated. The authors mentioned that targeting lncRNA can serve as a new opportunity, but how ? There is no detailed explanation how it can targeted ?

Further, they mentioned the direction of future research to search for suitable lncRNA molecules. But the authors quote is confusing—does the author want to state finding the suitable lncRNA that regulates ferroptosis or finding molecules that can inhibit the function of lncRNA ?

Further, there is no clear explanation of the mechanism by which the lncRNA controls ferroptosis, apart from vaguely mentioning the regulation of SLC7A11 and GPX4. Table 1 is organized well, but in the main text, the authors should provide their point of view independently on how they can regulate ferropotosis. Its well known that most of the mechanisms of the lncRNA are mediated by its structures so I recommend the authors discuss on that aspect. Kindly check some recent related articles. Also you may want to consider this article on targeting lncRNA - https://pubmed.ncbi.nlm.nih.gov/37225982/

The figures, in essence, are not providing a complete understanding of the mechanism. For instance, many key aspects of the figures are not explained. The authors should consider improving the figures and providing more information on the legends.

In conclusion, I recommend the authors elaborate on the possible mechanism of the lncRNAs in ferroptosis in their point of view and discuss on the gaps/challenges associated with identifying the links and how to address them. Further, they should highlight the novelty in targeting those lncRNAs linked to ferroptosis and identifying candidates using HTS or so.

Experimental design

no comment

Validity of the findings

no comment

---

## Round 0.2 · accepted · Accept

All the issues indicated by the reviewers were adequately addressed, and the revised manuscript is acceptable now.

Reviewer 1 ·

Basic reporting

All my concerns have been addressed.

Experimental design

All my concerns have been addressed.

Validity of the findings

All my concerns have been addressed.

Reviewer 2 ·

Basic reporting

The authors have now revised the manuscript accordingly. I recommend it for publication.

Experimental design

No comment

Validity of the findings

No comment